# Contextualising the developability risk of antibodies with lambda light chains using enhanced therapeutic antibody profiling

Matthew I. J. Raybould[1], Oliver M. Turnbull [1], Annabel Suter[1], Bora Guloglu[1] & Charlotte M. Deane [1✉]

Antibodies with lambda light chains ($\lambda$-antibodies) are generally considered to be less developable than those with kappa light chains ($\kappa$-antibodies). Though this hypothesis has not been formally established, it has led to substantial systematic biases in drug discovery pipelines and thus contributed to kappa dominance amongst clinical-stage therapeutics. However, the identification of increasing numbers of epitopes preferentially engaged by $\lambda$-antibodies shows there is a functional cost to neglecting to consider them as potential lead candidates. Here, we update our Therapeutic Antibody Profiler (TAP) tool to use the latest data and machine learning-based structure prediction, and apply it to evaluate developability risk profiles for $\kappa$-antibodies and $\lambda$-antibodies based on their surface physicochemical properties. We find that while human $\lambda$-antibodies on average have a higher risk of developability issues than $\kappa$-antibodies, a sizeable proportion are assigned lower-risk profiles by TAP and should represent more tractable candidates for therapeutic development. Through a comparative analysis of the low- and high-risk populations, we highlight opportunities for strategic design that TAP suggests would enrich for more developable $\lambda$-antibodies. Overall, we provide context to the differing developability of $\kappa$- and $\lambda$-antibodies, enabling a rational approach to incorporate more diversity into the initial pool of immunotherapeutic candidates.

[1] Oxford Protein Informatics Group, Department of Statistics, University of Oxford, 24-29 St Giles', Oxford OX1 3LB, UK. ✉email: deane@stats.ox.ac.uk

Antibodies are the dominant category of biotherapeutics; more than 140 therapeutic antibodies have now been approved by regulators with over 550 currently active in clinical trials[1,2]. Their popularity is tied to their use by natural immune systems and their ability to achieve high affinity/specificity for seemingly any targeted pathogen (antigen), enabling its selective eradication[3].

Conventional antibodies are dimeric, comprise two identical heavy and light chains and accomplish precise antigen recognition through two dedicated antigen binding sites, termed the variable regions (Fvs). These Fvs are identical and structurally/chemically intricate, containing six proximal complementarity-determining region (CDR) loops — three on the variable domain of the heavy chain (VH, CDRH1-3) and three on the variable domain of the light chain (VL, CDRL1-3).

A large portion of the VH sequence derives from the recombination of a heavy V, D, and J gene, while most of the VL sequence is analogously the product of recombination of a light V and J gene. These heavy and light chain immunoglobulin germline genes are encoded at different loci across the chromosomes. For example, in humans, the heavy chain V, D, and J genes (IGHV, IGHD, IGHJ) lie solely on chromosome 14, while light chain V and J genes exist at two loci; a kappa ($\kappa$, IGKV and IGKJ) locus on chromosome 2, and a lambda ($\lambda$, IGLV and IGLJ) locus on chromosome 22[4,5].

Within each locus, different V(D)J genes recombine to create a considerable baseline diversity in both the VH and VL sequence[6]. Nucleotide insertions/deletions in the junction region between genes (which falls within the CDR3 loops) further contribute to exceptional VH and VL sequence diversification. Pairing of the recombined heavy and light chains then adds an additional combinatorial diversity; in this manuscript we term antibodies containing a $\kappa$ light chain as $\kappa$-antibodies, and those containing a $\lambda$ light chain as $\lambda$-antibodies. Finally, antibody sequence diversity is magnified through somatic hypermutation during an immune response. This process is often artificially mimicked during therapeutic development through in vitro affinity maturation/engineering.

Although the VH sequence is more diversified, the VL sequence is often critical to an antibody's function. For example, it has been observed in different toxin, virus, and vaccine response contexts that $\kappa$- and $\lambda$-antibodies are expressed in characteristic proportions with restricted usages, and that they tend to have different antigen specificities[7]. Amongst the thousands of anti-coronavirus antibodies independently isolated throughout the pandemic, the same VL germline genes have been frequently observed amongst antibodies with a high confidence of engaging the same epitope[8,9]. This link between VL germline genes and function has recently been shown to apply more generally, as evidenced by Jaffe et al. who found light chain coherence of memory B-cell compartments[10], and by Shrock et al. who identified the presence of germline amino acid-binding motifs — many of which lie in the VL sequence[11]. Together, these phenomena are likely by-products of the documented sequence[11–14] and structural[11,15–17] differences between $\kappa$- and $\lambda$-VLs, which may have evolved to increase the efficacy of receptor editing[18], a process during which maturing BCRs can exchange their initial recombined $\kappa$ light chain for a $\lambda$ light chain to prevent autoreactivity.

Despite their functional utility, $\lambda$-antibodies are currently under-represented across clinical-stage therapeutic antibodies (CSTs). Of a set of 242 CSTs curated in 2019[19], all of which were designed for human application, only around 10% derived from $\lambda$-genes. By comparison, $\lambda$-antibodies are estimated to comprise roughly 35% of natural human repertoires[20,21].

The precise reasons behind the paucity of $\lambda$-antibodies in the clinic are unknown, but there are several probable origins. These include factors related to the dominant methods of therapeutic discovery[22,23], such as unintended selection bias in screening library design[24] and the higher $\kappa$:$\lambda$ ratios (up to 20:1) seen in mouse antibody repertoires[25].

There are also reports that suggest that $\lambda$-antibodies may fail more frequently than $\kappa$-antibodies to advance through pre-clinical development[26]. Several studies have identified that $\lambda$-VLs exhibit a higher average hydrophobicity than $\kappa$-VLs[12,13,18,19]; higher hydrophobicity suggests an increased propensity towards the formation of aggregates via the hydrophobic effect. This mechanism is understood to be the primary force driving light chain amyloidosis, where free light chains self-associate, and data suggest that $\lambda$-VLs prone to dimerisation outnumber $\kappa$-VLs[27].

This has earned $\lambda$-antibodies a reputation for poor developability that has fed back into systematic discovery biases, such as through the intentional development of $\kappa$-only screening libraries[28], or, when given a choice of progressing $\kappa$- or $\lambda$-antibodies to downstream lead optimisation, a tendency to prioritise the former. However, it is probable that a sizeable proportion of $\lambda$-antibodies are indeed developable, and that rational engineering could be used to make some more challenging $\lambda$-antibodies biophysically tractable[29]. In general, better distinction between more developable and less developable $\lambda$-antibodies should be applied to limit the degree to which we artificially restrict candidate diversity, and therefore targetable epitope space, during early stage discovery.

In 2019, we published the Therapeutic Antibody Profiler (TAP), a method for the computational developability assessment of lead candidates based on comparing their 3D biophysical properties to those of CSTs[19]. At the time, we had access to only 25 $\lambda$-based CST sequences and artificially-paired representations of natural human antibodies. Now, through dedicated efforts to track the sequences of CSTs as they are designated by the World Health Organisation (WHO)[1] and increased public availability of paired-chain natural antibody repertoires[30], we are able to more confidently profile the physicochemical properties of therapeutic and natively-expressed human $\lambda$-antibodies.

In this paper, we first improve TAP by incorporating ABodyBuilder2[31], a state-of-the-art deep-learning based antibody structure prediction method and highlight changes and robustness of the new guideline values. We then use our updated protocol to characterise developability-linked biophysical differences across CST and natural $\kappa$- and $\lambda$-antibodies. Finally, we probe the subset of red-flagging antibodies for recurrent features associated with extreme scores, and which may be avoided to derisk the incorporation of $\lambda$-antibodies into screening libraries.

Overall, our study provides an improved methodology for therapeutic antibody profiling and adds context to the developability of $\lambda$-antibodies, facilitating their selective consideration as leads during early-stage drug discovery.

## Results

**Curating datasets of therapeutic and natural antibodies.** We first curated the latest set of non-redundant, post Phase-I clinical stage therapeutics (CSTs) designated for use in humans from Thera-SAbDab[1] (25th January, 2023). We obtained 664 CST Fv sequences (the $CST_{all}$ dataset), compared to the 242 used in our previous analysis[19]. To obtain a reference set of natural human antibodies, we utilised the paired Observed Antibody Space (OAS) database[30], which tracks, cleans, and annotates single-cell antibody V(D)J sequencing datasets in the public domain. We curated all 79,759 non-redundant natively-paired human antibody sequences from paired OAS (25th January, 2023), which we term the $Nat_{all}$ dataset. This compares to datasets of between 14,000–19,000 artificially-paired human antibody sequences used in our previous analysis[19].

**Table 1 Latest TAP thresholds based on ABodyBuilder1 or ABodyBuilder2 models.**

| TAP Property | ABodyBuilder1 | | ABodyBuilder2 | |
|---|---|---|---|---|
| | Amber Flag Region | Red Flag Region | Amber Flag Region | Red Flag Region |
| $L_{tot}$ | 37 to 42 | <37 | 37 to 42 | <37 |
| | 55 to 63 | >63 | 55 to 63 | >63 |
| PSH | 90.88 to 114.61 | <90.88 | 95.58 to 110.11 | <95.58 |
| | 174.26 to 219.20 | >219.20 | 168.06 to 201.59 | >201.59 |
| PPC | 1.27 to 4.21 | >4.21 | 1.32 to 4.22 | >4.22 |
| PNC | 1.93 to 4.36 | >4.36 | 2.00 to 4.42 | >4.42 |
| SFvCSP | −30.00 to −6.00 | <−30.00 | −30.60 to −6.00 | <−30.60 |

Flagging regions across the five TAP developability metrics calculated over the $CST_{all}$ dataset (See Methods), when therapeutics are modeled by ABodyBuilder1 or by ABodyBuilder2. As ABodyBuilder1 could not find sufficiently homologous templates to produce a model for Basiliximab, Iscalimab, and Netakimab, its statistics are calculated over 661/664 CSTs. Amber and red flag percentiles were set as per Raybould et al. 2019[19]. $L_{tot}$ Total CDR Length, *PSH* Patches of Surface Hydrophobicity metric, *PPC* Patches of Positive Charge metric, *PNC* Patches of Negative Charge metric, *SFvCSP* Structural Fv Charge Symmetry Parameter.

**Benchmarking a new TAP modeling protocol**. The original Therapeutic Antibody Profiler used the homology modeling tool ABodyBuilder[32] (ABodyBuilder1, for clarity) for antibody structural modeling. In 2018, this was the state-of-the-art tool for high-throughput antibody modeling. However, recent advances in deep learning have yielded several pretrained ab initio structure prediction architectures that can be applied or adapted to the task of rapid antibody/CDR loop modeling[31,33–35]. Their average performance has been shown to be consistently higher than that of homology-based antibody modeling methods. Since better models of antibodies should improve the reliability of our developability guidelines, we explored the case for updating our TAP protocol to use a more recent machine learning-based tool (ABodyBuilder2[31]) for 3D structural modeling. We selected ABodyBuilder2 as it has been shown to outperform other antibody-specific modelling methods[31], while being competitive with AlphaFold Multimer[33] at orders of magnitude faster modelling rates.

We first confirmed that ABodyBuilder2's improved general performance translates specifically to CSTs, observing increased backbone and side chain modeling accuracy relative to ABody-Builder1 across a set of recently-solved therapeutics (Supplementary Note 1, Supplementary Tables 1–2). This motivated us to formally adopt ABodyBuilder2 as the tool for 3D modeling prior to computation of the TAP metrics.

We next analysed the impact of using ABodyBuilder2 *versus* ABodyBuiler1 for structural modeling on the TAP developability guidelines calculated across the $CST_{all}$ set. We measure this based on their impact on the amber and red flagging thresholds — characteristic percentile values used to demark the extrema of each TAP property distribution linked to poor developability[19].

For reference, amber flags for Total CDR Length ($L_{tot}$) or Patches of Surface Hydrophobicity (PSH) are assigned to scores in the 0th-5th or 95th-100th percentiles relative to CSTs, while red flags are assigned if the $L_{tot}$/PSH score falls below the 0th or above the 100th percentile. Amber flags for Patches of Positive Charge (PPC) or Patches of Negative Charge (PNC) are assigned if a score falls in the 95th–100th percentile range relative to CSTs, and red flags are assigned to scores above the 100th percentile. Finally, amber flags are assigned for the Structural Fv Charge Symmetry Parameter (SFvCSP) metric if the score falls between the 0th-5th percentile values relative to CSTs, while red flags are assigned to scores below the 0th percentile.

The ABodyBuilder2-modeled $CST_{all}$ flagging thresholds show high similarity to those obtained by ABodyBuilder1 (Table 1, Supplementary Fig. 1). There is, however, some evidence of a systematic bias associated with the different modeling protocol. Comparing the amber flag thresholds (less volatile than red flag thresholds as they capture 5% of the data) shows that

ABodyBuilder2-modeled CSTs have lower PSH scores than ABodyBuilder1-modeled CSTs. This drop in PSH score is consistent with ABodyBuilder2's more accurate modeling; we found in our original TAP paper that PSH values calculated over solved crystal structures (theoretically perfect predictions) were lower on average (a difference of *c.* 10) than those calculated over ABodyBuilder1 models of the same CSTs[19]. This emphasises the need to use the same modelling tool for setting the guidelines and evaluating new candidates.

**Testing the robustness of the TAP developability guidelines**. We then probed the robustness of our developability guidelines to various perturbations. TAP values calculated on the subset of CSTs modeled with higher certainty should be more reliable. Model confidence can be estimated through the frame-aligned prediction error (FAPE) metric, a property minimised as part of the ABodyBuilder2 loss function that can be interpreted as a measure of backbone prediction uncertainty for each residue[31]. To investigate the impact of FAPE-based confidence filtering on our guidelines, we first determined an appropriate CDRH3 root-mean squared predicted error threshold that would filter out the least-confidently modeled CDRH3s (1.31 Å, see Methods for the derivation), then calculated our developability guidelines based only on the subset of most confident CST predictions (the $CST_{conf}$ set, see Supplementary Fig. 2, Supplementary Table 3). This set of generally higher quality models provides a more accurate reference set of physicochemical distributions, which, if they were to differ substantially from the general set, would imply that ABodyBuilder2's model accuracy has a systematic impact on the aggregate guidelines set over all CSTs. Overall, the guidelines derived from only the most confident models aligned closely with those set over all CSTs, suggesting that the new TAP guidelines are robust to the variable prediction accuracy of ABodyBuilder2.

Next, we examined the effect of ABodyBuilder2's non-deterministic side chain modeling to explore statistically how side chain conformational uncertainty impacts the guidelines. We ran the TAP protocol three times per CST and investigated the consistency of structure-dependent TAP metrics for each CST (Supplementary Fig. 3). The results for all metrics were all highly consistent between runs. PPC, PNC, and SFvCSP values were the most consistent, with Pearson's coefficient values in the range of 0.993–0.996. Due to their sensitivity to structural variations in any CDR vicinity residue, we expected the PSH values to be more susceptible to inter-run fluctuations. This was borne out, however PSH remained strongly correlated between two independent modeling runs ($\rho$: 0.945), and even more strongly correlated between one run and the mean of three independent runs ($\rho$: 0.981). The proportions of flagging inconsistencies (instances where a CST would be flagged for that property based on one

TAP run but not based on three repeats), were as follows: PSH (lower): 3.31%, PSH (upper): 1.81%, PSH (overall): 5.12%, PPC: 0.30%, PNC: 0.30%, SFvCSP: 0.75%.

To capture the absolute variability of scores across repeats, we evaluated for each metric/CST the variance across the three runs and averaged these values on a per metric basis across the $CST_{all}$ dataset. The mean PSH variance was 10.53 while the mean PPC, PNC, and SFvCSP variances were below 1 (Supplementary Table 4). When values from three runs were amalgamated to establish aggregate TAP guidelines, this translated to a very small variation in threshold values from those obtained based on a single model of each CST (Supplementary Table 5).

In addition to this statistical sampling of energy-minimised side chain conformations, we also evaluated the variation in TAP scores calculated over the course of molecular dynamics simulations; incorporation of dynamics in guideline evaluation was suggested in a recent study on computational developability prediction[36]. We selected 14 case study CSTs, seven of which had solved Fv coordinates in the ABodyBuilder2 training set and seven of which did not; for details of the molecular dynamics simulation and TAP calculations, see Methods.

The profiles for each of the four structure-dependent TAP metrics are shown in the Supplementary Information (Supplementary Figs. 4–7). The mean value of the TAP properties over the course of the simulation showed good agreement with an ensemble of three TAP predictions on the static Fv models output directly by ABodyBuilder2. Based on a paradigm where if a developability flag is raised on any of the repeat calculations we consider the antibody formally flagged for that property, the ensemble of direct ABodyBuilder2 outputs agreed with the flag assigned to the simulation mean for 12/14 calculations for PSH, 13/14 calculations for PPC, and 14/14 calculations for PNC and SFvCSP. Furthermore, we tested whether three ABodyBuilder2 modelling runs were sufficient to explore the diversity of side chain conformations by doubling to six runs and comparing the results. Based on an analogous ensemble paradigm, the agreement remained the same. However, there was some evidence that the additional runs helped to improve statistical consensus with the simulation-mean flag (Supplementary Table 6). For example, Simaravibart and Regdanvimab - which were assigned flags for PSH based on the molecular dynamics - flagged in a higher proportion of the six runs than the first three (1/3 *vs.* 3/6 runs, and 2/3 *vs.* 5/6 runs, respectively).

**TAP metric profiles over time and by development status**. Finally, we investigated the impact on our metric distributions of filtering our CSTs by metadata properties.

To assess the properties of CSTs over time, we split the set by the year they were given a proposed WHO International Non-proprietary Name, yielding 356 named between 1987–2017 and 308 named between 2018 and the present day. Comparing their TAP property distributions (Supplementary Fig. 8) indicates that while their charge metrics are similar, the amber and red flag thresholds of the $L_{tot}$ and PSH properties have shifted to more extreme values at both tails, suggesting an increased willingness to push CST design into new property spaces and perhaps reflecting formulation advances able to accommodate more extreme physicochemical properties.

Recent studies have suggested that developability guidelines may be better derived from marketed therapeutics[36,37]; we evaluated our TAP distributions for the subsets of CSTs in Phase-II (341), Phase-III (141), or in Preregistration/Approved (182), however observed no clear trend in their properties along the clinical progression axis (Supplementary Fig. 9). Equally, we saw little difference in the properties of CSTs known to be in

active development or that completed the development pipeline *versus* CSTs whose campaigns were terminated before reaching approval (Supplementary Fig. 10). These observations are consistent with the principle that CSTs with unmanageable developability issues do not tend to progress past pre-clinical/Phase-I development, and that decisions to terminate campaigns at later clinical stages are often attributable to other causes.

**Updated comparison of CSTs to natural human antibodies**. A key biotechnological advance in recent years has been the advent of high-throughput paired B-cell receptor (BCR) sequencing[38]. Publicly available paired antibody sequences are increasingly abundant[30] and provide a higher fidelity comparison set than the artificially-paired natural single chain reads we used in previous repertoire characterisation work[19,39]. These samples, coupled with the availability of 2.75 times as many CSTs and a more accurate/versatile modeling protocol, provides an ideal opportunity to revisit prior analyses and explore whether we observe similar trends in the biophysical properties of therapeutics and natural antibodies.

We calculated the TAP profiles for our new curated datasets of naturally-paired human sequences ($Nat_{all}$, $N = 88,274$) and CSTs ($CST_{all}$, $N = 664$). The patterns of the distributions aligned with our findings in the original paper (Fig. 1a–e). CSTs and natural human antibodies adopted similar PPC, PNC, and SFvCSP distributions, but natural antibodies were even more enriched at longer CDRs and higher PSH scores than observed previously (30.16% and 24.23% fall above the upper amber flag thresholds set by the top-5% of CSTs, respectively). To ensure the length bias was not the sole driver of higher PSH scores, we plotted the $L_{tot}$ against the PSH score for the $Nat_{all}$ and $CST_{all}$ datasets (Fig. 1f). While almost all the natural antibodies found at extreme $L_{tot}$ values flag for PSH, so too do a disproportionate number of natural antibodies at more moderate CDR lengths, even down to the smallest recorded $L_{tot}$ value.

To further test the robustness of these conclusions, we then restricted our analysis to a confidence-filtered subset of ABodyBuilder2 models of the CSTs ($CST_{conf}$, $N = 510$) and the natural data ($Nat_{conf}$, $N = 30,402$), generated using the FAPE threshold benchmarked on CSTs (see Methods). In these sets a much smaller fraction of natural antibodies survived the filtering cut-off (~38% of $Nat_{all}$ *versu* ~75% of $CST_{all}$), likely due to the fact that natural antibodies sample longer CDRH3 lengths — which are both more conformationally diverse and harder to crystallise — as well as the general under-representation of natural antibodies in the Protein Data Bank[40], on which ABodyBuilder2 is trained.

The resulting $CST_{conf}$ and $Nat_{conf}$ TAP distributions show analogous relative positioning to our original results[19], with CSTs occupying shorter $L_{tot}$ and smaller PSH values than natural antibodies, but having similar charge characteristics (Supplementary Fig. 2). Quantitatively, over 16% and over 18% of $Nat_{conf}$ antibodies surpassed the $L_{tot}$ and PSH upper amber thresholds set by the $CST_{conf}$ set (compared with ~30% and ~24% on the $Nat_{all}$ set, respectively). The large reduction in the number of natural antibodies flagging for $L_{tot}$ confirms that antibodies with longer CDR loops are modeled with lower confidence. The smaller percentage reduction in natural antibodies flagging for PSH reflects the increased tendency for natural antibodies of all lengths to occupy higher PSH values.

In summary, our investigations strengthen the evidence that CSTs and natural antibodies differ in their CDR length and surface hydrophobicity properties.

**Using the new TAP protocol to explore λ-antibody developability**. We then used our updated TAP protocol to explore the relative developability of κ- and λ-antibodies.

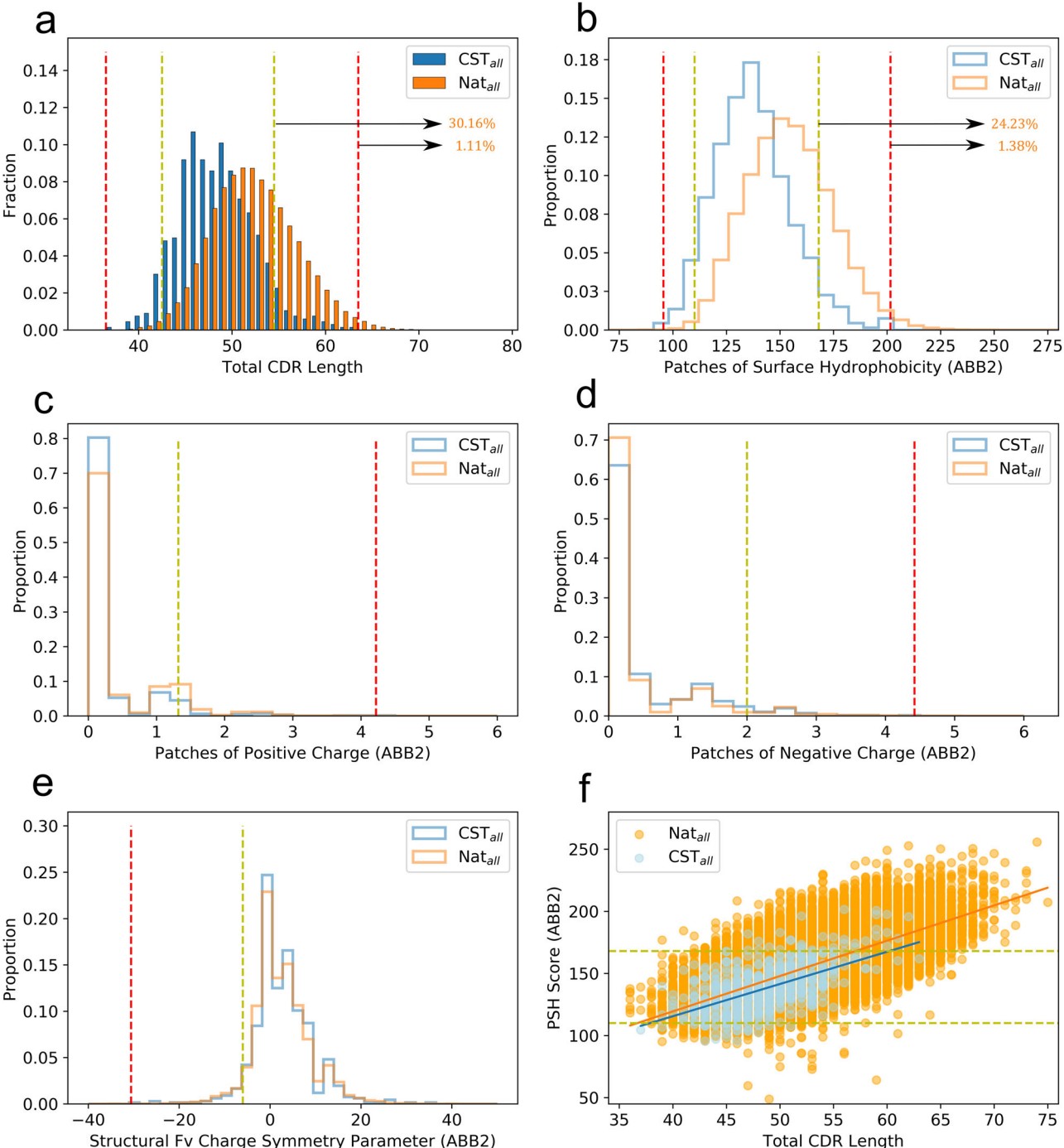

**Fig. 1 TAP metric distributions based on ABodyBuilder2 models of the latest therapeutic data. a–e** The five TAP developability metric distributions set by using ABodyBuilder2[31] (ABB2) to model the curated $CST_{all}$ (blue) and $Nat_{all}$ (orange) datasets. **f** A plot displaying the trend between $L_{tot}$ and PSH Score for both datasets. Amber and red flag thresholds are shown with dashed lines of corresponding color. The percentages of $Nat_{all}$ antibodies lying above the $L_{tot}$ and PSH upper thresholds are highlighted.

We examined the growth trends of $\kappa$- and $\lambda$-CSTs. Plotting their numbers over time reveals distinct patterns in usage (Fig. 2). For example, while novel $\kappa$-CST Fvs have been continuously released in double-digit quantities per year since 2007, new $\lambda$-CST Fvs only reached this level in 2018. In 2019, for example, the industry developed 53 new $\kappa$-CST Fvs, but only 10 new $\lambda$-CSTs.

This lag has led to a disparity in the abundance of $\kappa$- and $\lambda$-CSTs. As of January 2023, Thera-SAbDab contained 576 non-redundant $\kappa$-CST Fvs (86.7%) and 88 non-redundant $\lambda$-CST Fvs (13.3%); far below the relative abundance of human $\lambda$-antibodies

(30–35%[20,21]). However, prior to the disruption of the pandemic, there were signs of an upwards growth trend in $\lambda$-CSTs (Fig. 2). There is evidence to suggest this is driven by the propensity of $\lambda$-VLs to bind different targets/epitopes to $\kappa$-VLs; amongst the therapeutics designated by the WHO since 2022, six $\lambda$-antibodies (Acimtamig, Firastotug, Golocdacimig, Temtokibart, Tolevibart, and Zinlivimab) are first-in-class clinical candidates against novel antigen targets or epitopes (FCGR3A, HHV gB AD, OLR1, IL22RA1, HPV Envelope Protein, and the HIV-1 gp120 V3 epitope, respectively[1]). Overall, the 88 sequence non-redundant

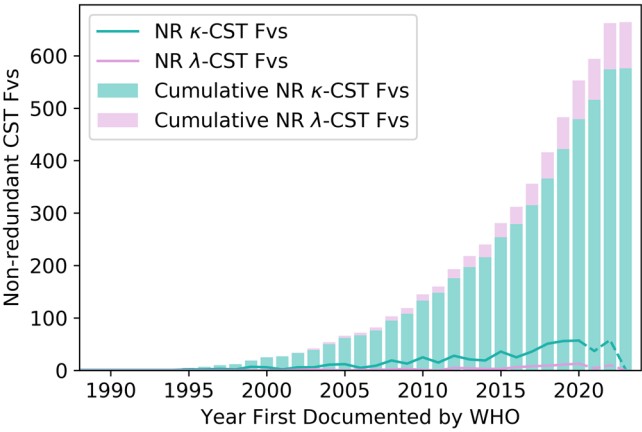

**Fig. 2 Progression of kappa and lambda therapeutic antibodies into late-stage clinical trials, by year.** Tracking the numbers of 100% sequence non-redundant $\kappa$ and $\lambda$ variable regions (Fvs) across post Phase-I sequence non-redundant clinical stage therapeutics (NR CSTs) from 1988 to 2023. The x-axis reflects the year in which each CST was granted a proposed International Non-proprietary Name (INN) by the World Health Organisation (WHO). Cumulative totals are shown through a stacked bar chart, while year-by-year totals are shown in the line graph. 2021–2023 are shown in dashed lines; these totals will likely change markedly once more therapeutics first reported in these years have had time to advance past Phase-I Clinical Trials.

$\lambda$-CSTs now in Thera-SAbDab represents a 250% increase on the 25 $\lambda$-CSTs we had access to when developing our original guidelines.

**TAP distributions across $\kappa$- and $\lambda$-antibodies**. We studied the CST biophysical property distributions for the two types of light chain (Fig. 3a–e; kappa $N = 576$, lambda $N = 88$).

$\lambda$-CSTs disproportionately amber flag at the upper extrema of the $L_{tot}$ (3.1% $\kappa$, 27.27% $\lambda$) and PSH (2.7% $\kappa$, 21.1% $\lambda$) distributions, and, to a lesser extent, for PPC (4.1% $\kappa$, 11.8% $\lambda$). In contrast, $\kappa$-CSTs predominate in the lower extrema of the $L_{tot}$ (10.4% $\kappa$, 1.3% $\lambda$) and PSH (5.8% $\kappa$, 0% $\lambda$) distributions. While the relative proportions in the flagging region for the SFvCSP score are similar, only $\kappa$-CSTs occupy the most extreme values (below $-15$).

As PSH values correlate to some extent with $L_{tot}$, we checked whether the preponderance of $\lambda$-CSTs at high PSH values was simply a by-product of length (Fig. 3f). Our results indicate that $\lambda$-CSTs are not noticeably more driven towards high PSH scores by longer CDR lengths than $\kappa$-CSTs are, with $\lambda$-CSTs having higher average PSH scores than $\kappa$-CSTs at every sampled $L_{tot}$ value.

The observation that $\lambda$-CSTs sit at such longer average $L_{tot}$ values than $\kappa$-CSTs was surprising. While $\lambda$-CDRL3s are known to be longer on average than their $\kappa$ equivalents[18], this alone cannot explain the shift. Instead, for this dataset, the disparity is also driven by biased pairing of $\lambda$-VLs with VH chains with longer average CDRH3 lengths ($\mu_{\kappa\text{-CST, H3}}$: 12.53 ± 3.07, $\mu_{\lambda\text{-CST, H3}}$: 14.30 ± 3.87).

We then studied the biophysical property distributions for the natural human sets of $\kappa$- ($N = 44{,}420$) and $\lambda$-antibodies ($N = 35{,}341$; Fig. 4). On these datasets we found a much smaller difference in $L_{tot}$ scores between the natural $\kappa$-antibody and $\lambda$-antibody models than observed in the CSTs; an offset fully explained by CDRL3 length biases across the two types of light chain ($\mu_{\kappa\text{-Nat, L3}}$: 9.12 ± 0.8, $\mu_{\lambda\text{-Nat, L3}}$: 10.61 ± 1.03). This result is consistent with Townsend et al.[18] and provides strong evidence

that the longer CDRH3 lengths seen in $\lambda$-CSTs are due to a bias (e.g. species origin[19]) in therapeutic development.

Analogous to the CSTs, natural human $\lambda$ antibodies were disproportionately flagged for high PSH relative to human $\kappa$-antibodies, but both were flagged at an even higher rate: 11.26% of natural $\kappa$-antibodies and 40.52% of natural $\lambda$-antibodies flagged, relative to 1.91% and 26.14% of $\kappa$-CSTs and $\lambda$-CSTs, respectively.

Both $\kappa$-CSTs and $\lambda$-CSTs therefore occupy a lower-risk region of CDR length and PSH property space relative to natural antibodies, strengthening the findings from the original TAP paper[19] where we suggested that CSTs in general require more conservative values of these properties than natural antibodies to be amenable to therapeutic development. It also highlights the complexity of developability optimisation in drug discovery: improvements in the humanness of the antibodies in screening libraries can have the unintended byproduct of enhanced therapeutic aggregation risk, regardless of the genetics of the VL sequence.

The charge properties of the natural $\kappa$- and $\lambda$-antibodies can be found in Supplementary Fig. 11. The natural $\lambda$-antibodies also show an enhanced propensity for PPC values over 1 relative to their kappa equivalents, suggesting a natural origin for the disproportionate flagging of $\lambda$-CSTs for PPC.

**Residue positions associated with driving $\lambda$-antibodies towards high PSH scores**. Our analyses suggest that the structure-dependent property biases across $\lambda$-CSTs are inherited from natural trends, especially for the PSH score. We therefore examined the $\lambda$-CSTs to determine which features in the Fv tend to correlate with their high PSH scores, with a view to guiding rational engineering and library design.

We selected the upper red-flagging sets of natural $\kappa$- ($N = 134$) and $\lambda$-antibodies ($N = 968$) and decomposed the overall TAP PSH score into its pairwise-residue component parts. We investigate in more detail the top-20-most hydrophobic sequence-adjacent interactions, and top-20-most hydrophobic sequence non-adjacent interactions, across antibodies red-flagging for PSH.

We observed a broad diversity of VH (Supplementary Fig. 12) and VL (Supplementary Fig. 13) residues involved in driving extreme PSH scores, emphasising the challenge of finding molecular rules of thumb for antibody optimisation. However, the VH residue positions involved in elevating the PSH of either $\kappa$-antibodies or $\lambda$-antibodies were highly similar, suggesting minimal bias in the physicochemical properties of VH sequences associating with $\kappa$- or $\lambda$-VLs.

Of particular interest to antibody optimisation engineers are positions outside of the CDR regions, since mutations at these sites are less likely to impact antigen specificity. Amongst the dominant residues contributing to high PSH scores were $\kappa$-VL positions 1–3 (framework region L1) and 79–85 (framework region L3), and $\lambda$-VL positions 24–26 (framework region L1) and 71–72 (framework region L3); while not in the formal CDRs, these residues lie in the vicinity of the CDRs and may be serving to extend hydrophobic self-association surfaces.

As a case study, we investigated in more detail VL positions 24–26, which drive higher PSH scores in $\lambda$-antibodies but not $\kappa$-antibodies (Fig. 5a, Supplementary Fig. 14). The $\lambda$-antibodies exhibit a broader diversity of amino acids at these positions than $\kappa$-antibodies, although, with the exception of leucine at position 25 (Supplementary Fig. 14), do not exhibit particularly hydrophobic residues. However, we observed that over 90% of $\kappa$-antibodies have positively-charged residues at position 24, while $\lambda$-antibodies almost exclusively use smaller, less polar residues (Fig. 5a), which would be expected to be more accommodating of a

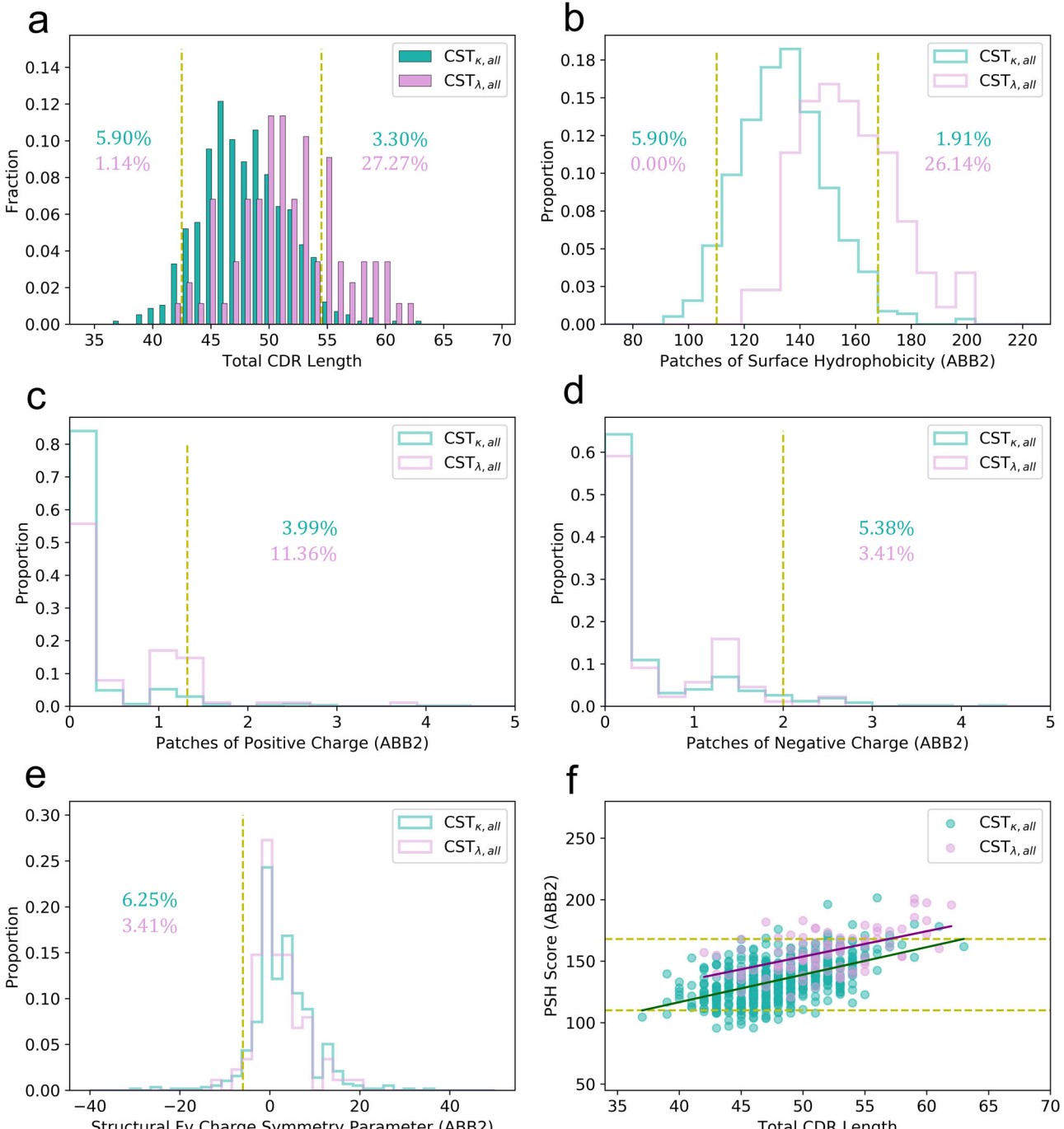

**Fig. 3 TAP properties of kappa and lambda therapeutic antibodies. a–e** Plots of the five TAP properties for the $\kappa$ (seagreen) and $\lambda$ (plum) CSTs, and (**f**) the trend of the Patches of Surface Hydrophobicity (PSH) score with $L_{tot}$, using ABodyBuilder2 for structural modeling (ABB2). Amber thresholds are set based on the 5th and/or 95th percentile values of the combined set of kappa and lambda CSTs. Percentage values reflect the proportions of the correspondingly coloured light-chain class of antibody within the amber-flagged region of each distribution.

hydrophobic self-association interface. Meanwhile, though serine is the most commonly observed residue at position 26 in $\lambda$-antibodies (and seen in 100% of $\kappa$-antibodies, Supplementary Fig. 14), threonine becomes by far the most prevalent residue amongst red-flagging $\lambda$-antibodies when position 26 is involved in the subset of most hydrophobic interactions (Fig. 5b). This is due both to its slightly higher intrinsic hydrophobicity and to more complex co-associations with other residues.

In summary, by decomposing TAP scores such as the PSH into pairwise residue components, we can elucidate the regions driving

high risk scores for individual or classes of antibodies and help to orient developability-motivated mutagenesis studies.

**$\lambda$-VL genes harbour characteristic risk profiles**. Associations of certain genes or gene families with PSH flagging propensity would offer a simple strategy to develop diverse but developable screening libraries, for example by incorporating only select lambda genes with a more moderate risk of poor developability. To investigate if such associations exist, we evaluated the gene

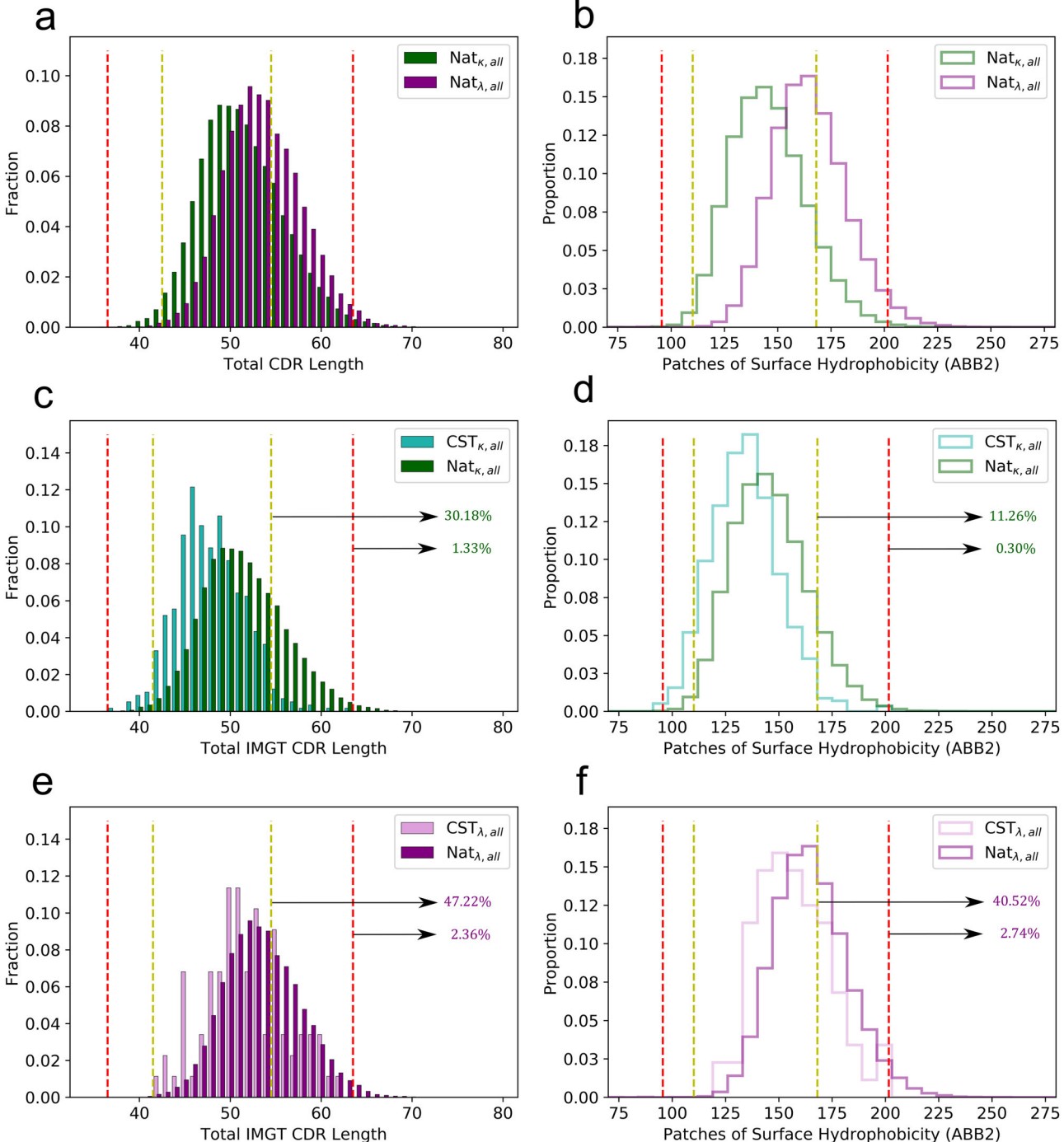

**Fig. 4 TAP properties of natural and therapeutic antibodies, split by light chain type.** Plots of the Total CDR Length (L$_{tot}$) and Patches of Surface Hydrophobicity (PSH) metric scores across different datasets split by light chain isotype. **a**, **b** L$_{tot}$ and PSH for the $\kappa$-Nat$_{all}$ and $\lambda$-Nat$_{all}$ subsets. **c**, **d** L$_{tot}$ and PSH for the $\kappa$-CST$_{all}$ and $\kappa$-Nat$_{all}$ subsets. **e**, **f** L$_{tot}$ and PSH for the $\lambda$-CST$_{all}$ and $\lambda$-Nat$_{all}$ subsets. Highlighted percentages show the proportions of $\kappa$-Nat$_{all}$ and $\lambda$-Nat$_{all}$ antibodies exceeding the upper TAP thresholds. ABB2: ABodyBuilder2 models.

usages of $\lambda$-antibodies that green-flagged ($N = 21,001$) or red-flagged ($N = 968$) for PSH (Fig. 5c, d). From a gene family perspective (Fig. 5c), *IGLV1* and *IGLV3* were associated with a lower PSH-mediated developability risk, while others such as *IGLV2*, were highly enriched amongst red-flagging $\lambda$-antibodies. *IGLV9* is almost exclusively found amongst flagging antibodies. These risk profiles are supported by the gene family usages across $\lambda$-CSTs (Supplementary Table 7): *IGLV1* and *IGLV3* are over-represented relative to their natural abundances, while *IGLV2* is under-represented, and no CST has yet derived from an *IGLV9* gene.

To study what features might be driving differential risk across families, we generated separate sequence logo plots of all *IGLV2* sequences and all non-*IGLV2* sequences (Supplementary Fig. 15). This highlighted positions that are commonly more hydrophobic among *IGLV2* antibodies. For example, position 57 is mostly valine with a trace of glycine in *IGLV2* antibodies, whereas it is predominantly asparagine or aspartate in antibodies from the other families. Similarly, position 3 is entirely hydrophobic across *IGLV2* antibodies but is found to be glutamate in roughly 1/3 of the other LV gene subgroups. Consistent with Fig. 5b, position 26

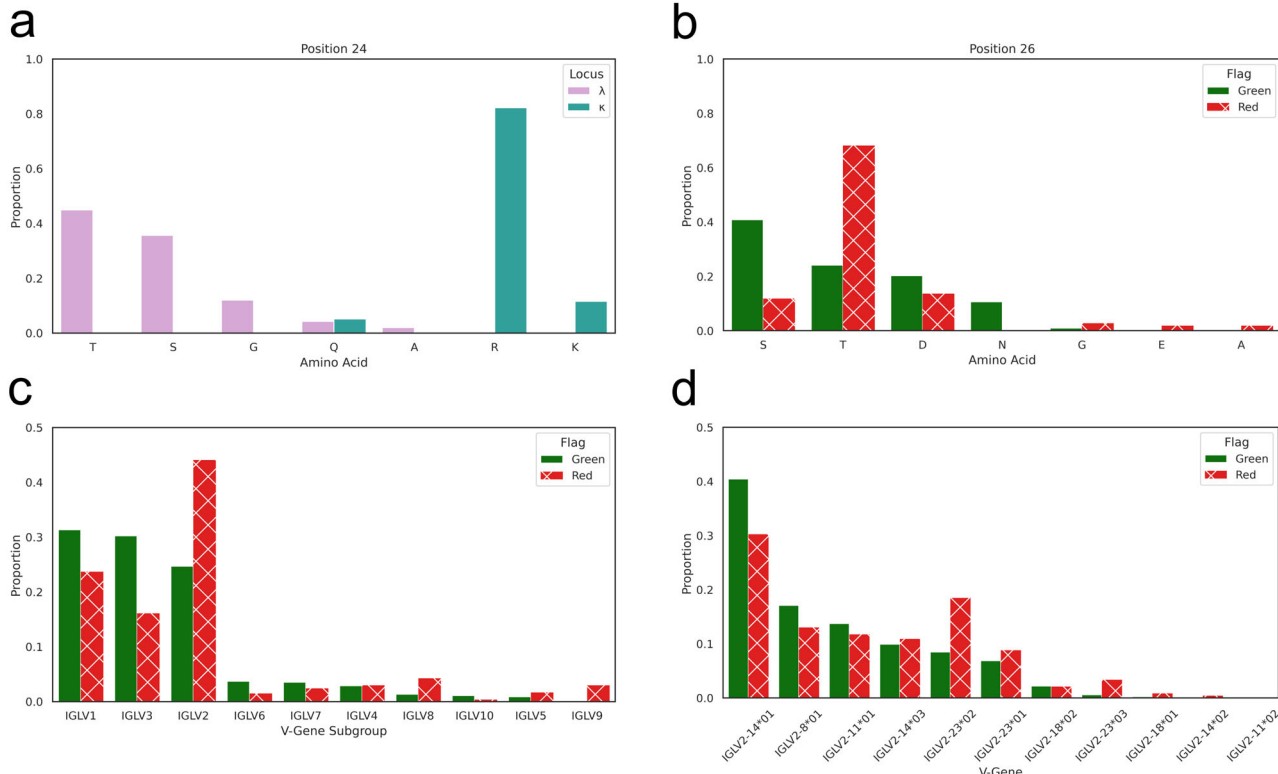

**Fig. 5 Features of green- and red-flagging kappa and lambda antibodies. a** Bar charts showing the amino acid usages at IMGT position 24 amongst natural λ-antibodies (plum) and natural κ-antibodies (seagreen). **b** Bar charts showing the amino acid usage at IMGT position 26 across natural λ-antibodies that are green-flagged for PSH, or that are red-flagged where position 26 is involved in the subset of most hydrophobic interactions. **c**, **d** Bar charts showing (**c**) the light V gene subgroup usages amongst natural λ-antibodies that are green-flagged or red-flagged for PSH, and (**d**) the *IGLV2* gene usages amongst natural λ-antibodies that are green-flagged or red-flagged for PSH.

is almost entirely threonine in *IGLV2* antibodies, while other families tend to use the less hydrophobic serine and highly polar residues such as asparagine and aspartate. Additionally, we observed that the CDRL1 loop, which typically bears a central motif containing hydrophobic residues, is frequently longer, and therefore more protruding, in *IGLV2* antibodies.

We then dissected the TAP PSH risk profiles for the *IGLV2* antibodies into profiles for individual genes (Fig. 5d). This demonstrated that the higher developability risk associated with the family is not shared evenly amongst its constituent genes: for example, *IGLV2-14*01* is found in a higher fraction of green-flagging antibodies than red-flagging antibodies, while every allele of *IGLV2-23* is associated with enhanced abundance among red-flagging λ-antibodies. Again, this is supported by gene usages across λ-CSTs: *IGLV2-14* is the dominant gene observed amongst the relatively small number of CSTs deriving from the *IGLV2* family (Supplementary Table 8).

Together these results suggest that a λ-antibody's germline V gene contributes substantially to its developability risk profile, and that TAP can be used to stratify lower from higher risk scaffolds.

## Discussion

In this paper, we benchmarked the latest machine learning-based antibody modeling technology for use in the Therapeutic Antibody Profiler[19].

We found that, while the precise guideline values we derived in 2019 have modulated slightly due to the availability of nearly three-times as many CST datapoints, the broad trends in property distribution between CSTs and natural antibodies have remained consistent; *i.e.* CSTs as a whole have shorter CDR loops and smaller patches of surface hydrophobicity, while their charge properties are highly similar. The patterns also hold when limited to the subset of higher-certainty models (based on ABody-Builder2's statistical heuristic[31]).

When split by year of designation by the WHO, new therapeutics are more frequently sampling the extremes of CDR and PSH property space, indicating that our definitions of druglikeness are likely to continue evolving over time. This phenomenon has also been observed in small molecule drug discovery, where several typical properties of today's drugs differ from the original trends documented by Lipinski et al.[41,42], and may be related to advances in developmental/formulation technologies.

On the other hand, we observe no obvious trends in the properties of post-Phase I active/approved therapeutics *versus* discontinued therapeutics, nor in therapeutics that have advanced to different clinical stages, suggesting that, at least in terms of the TAP properties, we would not expect predictive power to improve by only considering therapeutics that have advanced to later stages. Unfortunately, there remains a void in publicly available data on antibodies that failed pre-clinical evaluation due to poor developability, against which physicochemical property guideline thresholds could be benchmarked.

Due to ABodyBuilder2's modeling protocol, statistical uncertainty in side chain positioning can now be captured to some extent through repeat modeling and TAP calculations. Guidelines derived from repeat runs are almost identical to the guidelines derived from a single run per therapeutic, while mean variances of property values of the CST therapeutics are near-0 for charge metrics and only around 10 for the PSH metric. Variances on this order can lead to classification disparities across repeat runs

between adjacent boundaries (*i.e.* green/amber risk, or amber/red risk) but are extremely unlikely to result in the same antibody being assigned green risk and red risk for a given property.

Molecular dynamics simulations of a representative set of CST Fab models indicate that the flags assigned by an ensemble of repeat static ABodyBuilder2 predictions are highly consistent with simulation-average flags. Best agreement with simulation is obtained by considering an antibody to have flagged for a property if a flag is seen on any of the repeat runs. As running TAP multiple times takes a few minutes, several orders of magnitude faster than running molecular dynamics, repeat TAP calculations may offer a sensible strategy for high-throughput developability screening with consideration for side chain mobility.

We then used our new TAP protocol to investigate developability-linked property biases across $\kappa$- and $\lambda$-antibodies, exploiting the rise in both CST and paired-chain natural sequence data. $\lambda$-VLs have distinct epitope specificities to $\kappa$-VLs, driven by features such as locus-specific germline-encoded amino acid binding-motifs[11]. However $\lambda$-antibodies have been suggested to be less developable than $\kappa$-antibodies[26] and are heavily under-sampled amongst CSTs relative to their natural abundance. Therapeutic antibody profiling adds quantitative evidence that natural $\lambda$-antibodies are generally at higher risk of developablity issues, especially hydrophobicity-driven aggregation, than natural $\kappa$-antibodies. Indeed, the mean of the natural $\lambda$-antibody distribution sits just below the amber-flagging threshold; a substantial population of $\lambda$-antibodies are prone to being nudged into being flagged by, say, an affinity maturation pipeline based on unconstrained mutagenesis.

However, through a quantification of the risk of each $\lambda$-antibody, TAP profiles can now enable us to identify subpopulations expected to be more amenable to therapeutic development, and therefore to offer strategies towards augmenting the targetable epitope space through rational design. The observation of particular lambda gene associations with higher risk profiles, and a preliminary concomitant signal in the germline gene origin distributions of $\lambda$-CSTs that have so far progressed to the clinic, suggest that approaches such as family-holdout (e.g. all *IGLV2*) or gene-holdout (e.g. all *IGLV2-23*) libraries should enrich for $\lambda$-antibodies with lower expected developability risk. Alternatively, libraries could be constructed at a more granular level, incorporating more risk-prone genes but only when the associated sequence is considered by TAP to be lower risk. While sequence-by-sequence in vitro screening library design may still be a distant prospect, such approaches are gaining traction in the field of in silico library design[39,43].

The interpretability of the TAP metrics means they can be readily deconstructed to explore which regions of the CDR vicinity tend to contribute to higher developability risk. We show that the positions that contribute most to high risk scores in both $\kappa$- and $\lambda$-antibodies are diverse and distinct. Differences lie in the VL sequence itself rather than through any biases in the properties of their paired VH sequences. While preliminary, we note that some residues in the periphery of the CDR vicinity can help drive antibodies towards being red flagged; on a case-by-case basis, mutations to these regions may impact developability while being less likely to affect antibody specificity.

To date, TAP has primarily found use in industry for the early-stage removal of candidate antibodies more likely to suffer from developability issues. This increases the efficiency of drug discovery, but risks artificially constraining diversity, reinforcing current established property trends. We have shown how TAP, applied to identify more nuanced $\lambda$-VL residue and $\lambda$-gene associations with developability risk, could also guide selective consideration of a broader diversity of lead candidates and so enable access to a wider pool of epitopes.

## Methods

**Dataset curation**. The Therapeutic Structural Antibody Database (Thera-SAbDab) was downloaded on 25th January, 2023[1]. The entries were filtered for those designed for human application, that have reached at least Phase-II of clinical trials, and that have complete variable regions (Fvs, *i.e.* no single domain antibodies were carried forward). This set was then mined for sequence non-redundant Fv regions (at the level of 100% identity), to filter out biosimilars with no changes to the Fv and to reduce biases caused by the use of previously-developed monoclonal Fv domain sequences in new multispecific formats. This resulted in 664 non-redundant CST Fvs (the CST$_{all}$ dataset), of which 576 were $\kappa$-based (86.7%) and 88 were $\lambda$-based (13.3%). Thera-SAbDab light-gene locus labels were confirmed *via* alignment to the latest set of human and mouse IMGT V domains using ANARCI[44].

The sequence non-redundant (100% identity) Fv sequences of 88,274 natural human antibodies were retrieved from the Observed Antibody Space (OAS) database[30] (timestamp: 25th January 2023). These were filtered for sequences with complete CDRs[45], leaving 79,761 antibodies (the Nat$_{all}$ dataset): 44,420 (55.7%) $\kappa$-antibodies, 35,341 (44.3%) $\lambda$-antibodies.

**Benchmarking TAP modeling methods**. ABodyBuilder1[32] was run using template databases built on a copy of SAbDab[45,46] timestamped to 30th April 2022, and with a template sequence identity cut-off of 99% to ensure genuine models were produced[32]. ABodyBuilder2 was run using the pre-trained weights from the paper[35]. Relative performance to ABody-Builder1 was evaluated by root-mean-squared deviation (RMSD) by IMGT-defined region[47] and was calculated with an in-house script that first aligns each model structure to the ground truth structure based on the backbone atoms of the framework region of the investigated chain and then calculates the RMSD over the backbone atoms of the residues of the region (for heavy or light chains in the IMGT numbering scheme, CDR1: residues 27–38, CDR2: 56-65, CDR3: 105-117).

The classification of residues as solvent exposed or buried was based on an in-house implementation of the Shrake and Rupley algorithm[48], using a spherical probe of radius 1.4 Å. A residue 'X' was considered exposed if its solvent-accessible surface area (SASA) was ≥7.5% of its theoretical maximum value (based on the open-chain form of Alanine-X-Alanine)[19]. In accordance with the parametrisations of these theoretical maximum SASAs, all hydrogen atoms were stripped out of ABodyBuilder2 predictions prior to SASA calculations.

A threshold to filter out the least confident ABodyBuilder2 models was obtained by a two-step process. First, the 119/664 CST Fv domains for which 100% sequence identical X-ray crystal structures exist (identified using Thera-SAbDab metadata[1], Supplementary Table 2), were filtered out of the CST$_{all}$ dataset. The root-mean-squared predicted error for each remaining CST CDRH3 was then calculated as $\sqrt{\frac{\sum_{res(CDRH3)} PE^2}{L_{CDRH3}}}$, where PE represents backbone predicted error, res(CDRH3) represents the sum over all CDRH3 residues and L(CDRH3) represents the length of the CDRH3. Finally, the threshold was derived by evaluating the 75th percentile (1.31 Å). This filter was applied to retain the 510 most confidently-modeled CSTs (the CST$_{conf}$ dataset), and applied to the Nat$_{all}$ dataset to derive the 30,402 most confidently modeled natural antibodies (the Nat$_{conf}$ subset).

**Running TAP on ABodyBuilder2 models**. Sets of CST and natural antibody Fv domains were run through TAP and their five computational developability metrics calculated (Total IMGT-defined[47] CDR Length [$L_{tot}$], Patches of Surface Hydrophobicity using the Kyte and Doolittle scale [PSH], Patches of Surface Positive Charge [PPC], Patches of Surface Negative Charge [PNC] and Structural Fv Charge Symmetry Parameter [SFvCSP])[19]. PSH, PPC, and PNC metrics were calculated across the CDR vicinity (IMGT-defined CDR residues ±2 on each side plus any other surface exposed residue within 4.5 Å of one of these residues). Throughout the work, amber and red thresholds and were set at the percentile values suggested in the original paper[19].

Whenever the properties of $\kappa$- and $\lambda$-antibodies where compared, threshold values were calculated from the CST$_{all}$ set (*i.e.* not evaluated separately by light chain type).

**Assessing TAP score variation over molecular dynamics trajectories.** 14 CST Fab regions were modelled by grafting the constant regions of their crystal structures (see Supplementary Table 9) onto the Fv models generated by ABodyBuilder2, obtaining the initial arrangement by aligning the crystal and model Fv backbones (full Fab regions were used instead of Fv regions based on the results of previous studies[49]). We then modelled-in missing residues in the constant region using MODELLER v10.2[50] and generated 10 models using the 'very slow' refinement setting, selecting the lowest energy model. All systems were prepared and simulations performed using OpenMM v7.7[51]. N-methyl groups were used to cap C-termini using an in-house script. Next, using pdbfixer[51], we protonated the models at a pH of 7.5, soaked them in truncated octahedral water boxes with a padding distance of 1 nm, and added sodium or chloride counter-ions to neutralise charges and then NaCl to an ionic strength of 150 mM. We parameterised the systems using the Amber14-SB forcefield[52] and modelled water molecules using the TIP3P-FB model[53]. Non-bonded interactions were calculated using the particle mesh Ewald method[54] using a cut-off of distance of 0.9 nm, with an error tolerance of $5\times10^{-4}$. Water molecules and heavy atom-hydrogen bonds were rigidified using the SETTLE[55] and SHAKE[56] algorithms, respectively. We used hydrogen mass repartitioning[57] to allow for 4 fs time steps. Simulations were run using the mixed-precision CUDA platform in OpenMM using the Middle Langevin Integrator with a friction coefficient of 1 ps$^{-1}$ and the Monte-Carlo Barostat set to 1 atm. We equilibrated systems using a multi-step protocol detailed in Supplementary Table 10. Following equilibration, we performed 200 ns of unrestrained simulation of the NPT ensemble at 300K, calculating TAP properties over the final 120 ns of each simulation, when all systems had reached relatively stable RMSD values from their initial coordinates (Supplementary Fig. 16). To estimate convergence, we aligned Fv regions on the starting structure using mdtraj v1.9.6[58] and calculated the RMSD of the Fv domains relative to the starting structure.

**Determining molecular correlates with poor developability.** Natural human antibodies lying above the red flag thresholds set by TAP across all CSTs were investigated for recurrent molecular patterns that contribute towards their high scores. The PSH scores for each antibody were split into components from sequence-adjacent residues and components from sequence non-adjacent residues, and these pairwise interactions were separately rank-ordered by hydrophobicity. Germline assignments for natural sequences were taken from the OAS Paired metadata[30], which derives from IgBlast[59] alignments of each nucleotide sequence to a recent set of human genes from the IMGT GeneDB[5]. Germline assignments for CSTs were evaluated using ANARCI[44] on amino acid sequences (allele predictions were ignored here due to the difficulty of accurately assigning alleles at the amino acid level). All percentage abundances of gene/gene family usages across $\lambda$-CSTs were calculated based the subset that mapped closest to human rather than mouse germlines.

**Visualisations**. All visualisations were made using open-source PyMOL or matplotlib version 3.5.2.

**Statistics and reproducibility**. All statistics were calculated using the numpy Python package (version 1.23.3). For fairness, the relative performance of ABodyBuilder1 and ABodyBuilder2 was benchmarked using CSTs whose structures were not available in the database or training set of the model, respectively. Additionally, TAP metrics were calculated across a redundancy-filtered set of CSTs to reduce bias caused by biosimilars with no changes to the Fv and by the use of the same variable domain sequence in multiple formats. We explored the impact of performing up to six independent ABodyBuilder2 modeling runs on TAP metric values, thresholds, and agreement with molecular dynamics simulations.

## Data availability

All crystal structures of antibodies were downloaded from SAbDab[45]. Numerical source data for all figures and tables is supplied as Supplementary Data 1. Additional supplementary files can be accessed on Zenodo (10.5281/zenodo.10357509), including the curated structures used for ABodyBuilder2 benchmarking and the ABodyBuilder2 models of all CSTs analysed in this study. ABodyBuilder2 models of natural paired-chain human antibodies were released in the Supplementary Materials of Abanades et al.[31,60].

## Code availability

The updated TAP protocol is available on our web application (https://opig.stats.ox.ac.uk/webapps/tap) and the source code is available under a free academic licence *via* the Vagrant Virtual Machine (https://process.innovation.ox.ac.uk/software/p/15303a/sabbox-academic/1) and Singularity container (https://process.innovation.ox.ac.uk/software/p/20120-a/sabbox-singularity-platform---academic-use/1) releases of our SAbDab-SAbPred codebase. The ABodyBuilder2 source code is available from GitHub (https://github.com/oxpig/ImmuneBuilder), with model weights used in this study available on Zenodo (https://doi.org/10.5281/zenodo.7258552).

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

## Acknowledgements

The authors would like to thank Dr. Sandeep Kumar (Boehringer Ingelheim) for critically reviewing the manuscript. This work was supported by a Postdoctoral Research grant funded by Boehringer Ingelheim (MR), and funding from the UK Engineering and Physical Sciences Research Council (OT, reference EP/S024093/1) and the Wellcome Trust (BG, reference 102164/Z/13/Z).

## Author contributions

M.R.: designed the research, performed the research, analysed data, wrote and edited the paper. O.T.: performed the research, analysed data, wrote and edited the paper. A.S.: performed the research, analysed data. B.G.: performed the research, analysed data, wrote and edited the paper. C.D.: designed the research, analysed data, wrote and edited the paper, supervised the research.

## Competing interests

The authors declare the following competing interests: C.D. discloses part-time employment by Exscientia plc and membership of the Scientific Advisory Board of Fusion Antibodies and AI proteins.
