## [Peer Review File · Communications Biology]

nature portfolio

Peer Review File

~~**Open Access** This file is licensed under a Creative Commons Attribution 4.0 International License, which permits use, sharing, adaptation, distribution and reproduction in any medium or format, as long as you give appropriate credit to~~

~~the original~~ author(s) and the source, provide a link to the Creative Commons license, and indicate if changes were made. In the cases where the authors are anonymous, such as is the case for the reports of anonymous peer reviewers, author attribution should be to 'Anonymous Referee' followed by a clear attribution to the source work. The images or other third party material in this file are included in the article's Creative Commons license, unless indicated otherwise in a credit line to the material. If material is not included in the article's Creative Commons license and your intended use is not permitted by statutory regulation or exceeds the permitted use, you will need to obtain permission directly from the copyright holder. To view a copy of this license, visit <http://creativecommons.org/licenses/by/4.0/>.

Reviewers' comments:

Reviewer #1 (Remarks to the Author):

Summary

Lambda antibodies (λ -antibodies) are often considered less promising than kappa antibodies (κ -antibodies) in drug development, leading to biases in clinical therapeutics. However, recent research suggests that λ -antibodies have untapped potential. The authors assessed the development risks of κ - and λ -antibodies based on their properties. In this paper, authors enhanced the Therapeutic Antibody Profiler (TAP) by integrating deep-learning-based antibody structure prediction tool ABodyBuilder2. Using this upgraded protocol, they analyzed the biophysical distinctions related to developability between clinical-stage therapeutic antibodies (CST) and naturally occurring κ - and λ -antibodies. Additionally, they investigated specific antibodies that raise concerns regarding their characteristics and seek common patterns associated with extreme scores. The authors found that while λ -antibodies generally have a higher risk, a significant portion presents lower risk profiles, making them viable candidates for therapy. This analysis provides insights into designing more developable λ -antibodies through an improved methodology for therapeutic antibody profiling. It also offers context regarding the developability of λ -antibodies. Interestingly, the authors also benchmarked TAP using molecular dynamics approaches.

Major comments

- The authors of the study conducted a comparative analysis of TAP properties for predicted antibody structures generated using non-deep learning-based ABodyBuilder1 and deep learning-based ABodyBuilder2.

 Notably, there was an absence of a comparative assessment with alternative deep learning-based methods such as IgFold – can the authors comment on that?

 Furthermore, can the authors comment on the consistency of TAP results when employing different structure prediction tools.

- The authors used a dataset consisting of 664 post-Phase-I clinical-stage therapeutic antibodies (CST). Can the authors comment on the availability of negative controls for such kind of studies? There was also an absence of information regarding the application of the Therapeutic Antibody Profiler (TAP) to the CST antibodies that did not successfully progress through clinical trials.

Minor comments

- Three repetitions may not provide a sufficiently comprehensive exploration of all potential side-chain conformations. Therefore, it is advisable to apply a greater number of modeling iterations. Or can the authors show that three repetitions are sufficient?

- Can the authors comment on the use or suitability of Therapeutic Antibody Profiler (TAP) to unpaired data?

Reviewer #2 (Remarks to the Author):

In this manuscript, the authors applied their updated model and protocols to a larger set of available antibodies, resulting in improved results. They then utilized this approach with lambda-antibodies to highlight their associated risk factors. Overall, this study represents a significant advancement in the field of therapeutic antibody development, emphasizing the crucial role of machine learning methods in rational antibody engineering.

However, I have outlined my concerns below and kindly request that the authors address them during the revision:

1. Given the existence of several new antibody structure predictors, why was ABodyBuilder2 exclusively selected?
2. What were the reasons behind ABodyBuilder's inability to generate models for Basiliximab,

Iscalimab, and Netakimab?

3. Could the authors provide an in-depth explanation of the distinctions among PPC, PNC, and SFvCSP using both versions of ABodyBuilder? Additionally, please rationalize the superiority of the newer version.

4. Could the authors offer supporting data to elucidate why downsampling CSTs with longer CDRH3s resulted in a more robust guideline?

5. In Fig. 1A and F, the terms "IMGT CDR Length" and "CDR Length" were both employed. Could the authors clarify whether these terms are interchangeable in this context? If so, it would be advisable to use just one term consistently throughout the manuscript.

6. While the authors state that CSTs and natural antibodies differ in key physicochemical properties, Fig. 1C - E seems to depict a significant degree of similarity between them. Could the authors provide explanations for this apparent contradiction?

Reviewer #3 (Remarks to the Author):

Raybould et al. have conducted comprehensive analyses of lambda and kappa properties among clinical-stage therapeutic and natural antibodies, identifying biophysical properties and positions/residues that associate with antibody risk profiles and developability. They build these analyses into their existing tool framework, alongside updates to their machine learning models. Specifically, through this implementation they expand the functionality of their "Therapeutic Antibody Profiler" software, now offering the potential to refine and expand the inclusion of candidate antibodies in initial stages of development – this represents a great resource for the community. The manuscript is thorough and very nicely written. Barring some minor editorial comments, I would be willing to recommend this manuscript for publication in Communications Biology.

Minor Comments

1) Page 2, paragraph 2: It would probably be good to describe the overall composition of the antibody first (i.e., is composed of two identical heavy and light chains).

2) Page 2, paragraph 3: "VH" and "VL" are "encoded" by many genes of different types, which reside in several loci across the genome. If you're referring to human in this instance, you should articulate the details, perhaps combining this description with that in the following paragraph.

3) Page 2, paragraph 3: V(D)J recombination is not "largely random". This is an antiquated description of this process and should be edited.

4) Page 2, Paragraph 4: Should specify that you're discussing human, and a reference or two might be useful.

5) Page 2, Introduction: Use of "VL" vs. "light chain" is inconsistent. (this is also true throughout the manuscript).

6) Page 2, "the same light chain gene origins": This needs to be clearer ("gene origins" is vague). Do you mean the same germline genes are used consistently?

7) To improve ease of reading (without needing to review other publications resources), it might be useful to give a little more description of the Nat database used from OAS. Just a very brief description of how that dataset is curated would suffice. This would help more naïve readers understand what is being incorporated into the analysis with CST antibodies.

We thank the editor and reviewers for their valuable comments and suggestions, which have significantly improved the quality of our manuscript. We supply a point-by-point response below, with reviewer comments in black, our reply in blue italics, and manuscript changes in purple (all textual changes are underlined).

Reviewer #1 (Remarks to the Author):

Summary

Lambda antibodies (λ -antibodies) are often considered less promising than kappa antibodies (κ -antibodies) in drug development, leading to biases in clinical therapeutics. However, recent research suggests that λ -antibodies have untapped potential. The authors assessed the development risks of κ - and λ -antibodies based on their properties. In this paper, authors enhanced the Therapeutic Antibody Profiler (TAP) by integrating deep-learning-based antibody structure prediction tool ABodyBuilder2. Using this upgraded protocol, they analyzed the biophysical distinctions related to developability between clinical-stage therapeutic antibodies (CST) and naturally occurring κ - and λ -antibodies. Additionally, they investigated specific antibodies that raise concerns regarding their characteristics and seek common patterns associated with extreme scores. The authors found that while λ -antibodies generally have a higher risk, a significant portion presents lower risk profiles, making them viable candidates for therapy. This analysis provides insights into designing more developable λ -antibodies through an improved methodology for therapeutic antibody profiling. It also offers context regarding the developability of λ -antibodies. Interestingly, the authors also benchmarked TAP using molecular dynamics approaches.

We thank the reviewer for their thorough reading of our manuscript.

Major comments

- The authors of the study conducted a comparative analysis of TAP properties for predicted antibody structures generated using non-deep learning-based ABodyBuilder1 and deep learning-based ABodyBuilder2.
 Notably, there was an absence of a comparative assessment with alternative deep learning-based methods such as IgFold – can the authors comment on that?

We thank the reviewer for this question, also highlighted by Reviewer 2 (comment 1). We selected ABodyBuilder2 as a recent benchmark (10.1038/s42003-023-04927-7) has shown that it outperforms other structure predictors such as IgFold and EquiFold, especially over the CDRH3 region, which is the most challenging to model. ABodyBuilder2's performance is on par with AlphaFold Multimer, however takes a fraction of the time to complete each model. This is crucial when generating models of tens of thousands of human antibodies as a reference set and is appropriate for preclinical drug discovery, when many clones remain potential candidates.

We have now clarified our rationale for selecting ABodyBuilder2 in the Results section:

*“The original Therapeutic Antibody Profiler used the homology modelling tool ABodyBuilder (10.1080/19420862.2016.12057; ‘ABodyBuilder1’, for clarity) for antibody structural modelling. In 2018, this was the state-of-the-art tool for high-throughput antibody modelling. However, recent advances in deep learning have yielded several pretrained *ab**

initio structure prediction architectures that can be applied or adapted to the task of rapid antibody/CDR loop modelling (10.1101/2021.10.04.463034, 10.1038/s41467-023-38063-x, (10.1093/bioinformatics/btac016, 10.1038/s42003-023-04927-7). Their average performance has been shown to be consistently higher than that of homology-based antibody modelling methods. Since better models of antibodies should improve the reliability of our developability guidelines, we explored the case for updating our TAP protocol to use a more recent machine learning-based tool (ABodyBuilder2 (10.1038/s42003-023-04927-7)) for 3D structural modelling. We selected ABodyBuilder2 as it has been shown to outperform other antibody-specific modelling methods (10.1038/s42003-023-04927-7), while being competitive with AlphaFold Multimer at orders of magnitude faster modelling rates.”

 Furthermore, can the authors comment on the consistency of TAP results when employing different structure prediction tools.

We thank the reviewer for this point, also highlighted by Reviewer 2 (comment 3). Regardless of whether ABodyBuilder1 or ABodyBuilder2 is used for modelling, the PSH, PPC, PNC, and SFvCSP metrics calculated over the same sets of CSTs yield very similar distributions; as we mentioned in the results “The ABodyBuilder2-modeled CST_{all} flagging thresholds show high similarity to those obtained by ABodyBuilder1 (Table 1).” This is not surprising, since we deliberately picked metrics that would not be too sensitive to slight inaccuracies/changes in structure. For example, passing a threshold residue surface exposure is sufficient for a residue to be included in the PSH, PPC, PNC, and SFvCSP calculation. We do not weight based on the degree of exposure or sum over every inter-atomic distance (features that are extremely volatile across runs and particularly prone to systematic differences between tools). Scatter plots for each TAP metric calculated over all CSTs modelled either by ABodyBuilder1 or ABodyBuilder2 are shown in Fig. R1 (the new Fig S1). They show considerable agreement between the two methods on which CSTs sit at the most extreme property values, with some variation which is to be expected when substantially different 3D structures are predicted. Overall, we note that while the trends in property values are medium to strong positive, they do not align with $x=y$, emphasising the need to be consistent in using the same tool for setting the guidelines and evaluating unseen candidates.

In principle, we should base our TAP guideline thresholds on 3D models generated with state-of-the-art accuracy, a point we highlighted in the Results section: “Since better models of antibodies should improve the reliability of our developability guidelines, we explored the case for updating our TAP protocol...”. We have demonstrated the improved accuracy of ABodyBuilder2 models over ABodyBuilder1 models (Table S1, and reference 10.1038/s42003-023-04927-7). Moreover, as we covered during the section of guideline benchmarking, ABodyBuilder2’s non-deterministic modelling allows us to sample multiple static side chain states, which provides better quantitative agreement with expensive molecular dynamics simulations than single models. In time, as improvements in high-throughput antibody modelling continue, the TAP metrics should be re-evaluated using the new technology. We note that any guidelines should always be derived using sets of models of CSTs generated using the same modelling algorithm, to mitigate against corruption due to systematic biases (10.1073/pnas.1810576116).

Fig R1. Scatterplots showing the degree of consistency in TAP metric values and thresholds over all CSTs when evaluated on an ABodyBuilder1 (x-axis) or ABodyBuilder2 (y-axis) model. Amber thresholds based on the 5th and/or 95th percentile values for each modelling tool are shown with dashed lines. A least-squares line of best fit for each metric is shown in black, with $x=y$ in grey.

>We have added a new Figure S1, scatterplots of PSH, PPC, PNC, and SFvCSP scores for all CSTs evaluated on ABodyBuilder1 (x-axis) or ABodyBuilder2 (y-axis) models.

>This new figure is referenced in “Benchmarking a New TAP Protocol”: “The ABodyBuilder2-modeled CST_{all} flagging thresholds show high similarity to those obtained by ABodyBuilder1 (Table 1, Fig. S1).

>We have added the following sentence to “Benchmarking a New TAP Protocol”: “This emphasises the need to use the same modelling tool for setting the guidelines and evaluating new candidates.”

- The authors used a dataset consisting of 664 post-Phase-I clinical-stage therapeutic antibodies (CST). Can the authors comment on the availability of negative controls for such kind of studies?

We thank the reviewer for identifying this omission. We are limited to learning from the positive class (i.e. antibodies with sufficiently good developability to advance past Phase-I in the clinic) as pharmaceutical companies must publicly disclose the amino acid sequence of their clinical-stage antibody to obtain an International Non-proprietary Name.

Unfortunately, despite the potential utility of the set, industry does not publicly disclose antibodies that failed to progress through the pre-clinical phase due to poor developability.

We have now clarified this in the discussion section of the paper.

“Unfortunately, there remains a void in publicly available data on antibodies that failed pre-clinical evaluation due to poor developability, against which physicochemical property guideline thresholds could be benchmarked.”

There was also an absence of information regarding the application of the Therapeutic Antibody Profiler (TAP) to the CST antibodies that did not successfully progress through clinical trials.

We thank the reviewer for this comment; we did consider whether CSTs that were still in active development or that were discontinued had different TAP metric profiles in Figure S9. We observed no trends of note, as per the sentence in the results section: “Equally, we saw little difference in the properties of CSTs known to be in active development or that completed the development pipeline versus CSTs whose campaigns were terminated before reaching approval (Fig. S9)”.

We have changed the title of this subsection for clarity “TAP metric profiles over time and by development status”.

Minor comments

- Three repetitions may not provide a sufficiently comprehensive exploration of all potential side-chain conformations. Therefore, it is advisable to apply a greater number of modeling iterations. Or can the authors show that three repetitions are sufficient?

We thank the reviewer for this good suggestion. To investigate this we double the number of modelling runs (from three to six) to investigate whether this changed any of the conclusions. The same CSTs that agreed with the simulation mean over three runs agreed with it over 6 runs (paradigm: a flag across any of the repeat TAP runs is enough to flag the CST for that property). However, we note some value from running more simulations: for Simaravibart and Regdanvimab, both of which flagged in the MD simulation, 1/3 and 2/3 TAP runs flagged for PSH. These ratios increased to 3/6 and 5/6 TAP runs, respectively, suggesting that additional repeats may yield better statistical consensus with the simulation mean flag.

We have therefore added the result to the manuscript:

“Furthermore, we tested whether three ABodyBuilder2 modelling runs were sufficient to explore the diversity of side chain conformations by doubling to six runs and comparing the results. Based on an analogous ensemble paradigm, the agreement remained the same. However, there was some evidence that the additional runs helped to improve statistical consensus with the simulation-mean flag (Table S6). For example, Simaravibart and Regdanvimab - which were assigned flags for PSH based on the molecular dynamics - flagged in a higher proportion of the six runs than the first three (1/3 vs. 3/6 runs, and 2/3 vs. 5/6 runs, respectively).”

We added a new table (Table S6) with the numerical data for all the runs to complement this analysis.

- Can the authors comment on the use or suitability of Therapeutic Antibody Profiler (TAP) to unpaired data?

We thank the reviewer for this question. We would not recommend that TAP is applied to unpaired antibody VH or VL sequences alone, as the region that would otherwise be occluded in the VH/VL interface would be falsely considered solvent exposed. There would also be the logistical challenge that ML-based antibody structure prediction tools require both heavy and light chain inputs to produce a model. However, TAP could be applied to “single chain” immune proteins that do not conventionally have a light chain partner such as nanobodies (VHHs), modelled using NanobodyBuilder2, for instance, if one could derive an appropriately large reference set.

--

Reviewer #2 (Remarks to the Author):

In this manuscript, the authors applied their updated model and protocols to a larger set of available antibodies, resulting in improved results. They then utilized this approach with lambda-antibodies to highlight their associated risk factors. Overall, this study represents a significant advancement in the field of therapeutic antibody development, emphasizing the crucial role of machine learning methods in rational antibody engineering.

We thank the reviewer for their kind words about our manuscript.

However, I have outlined my concerns below and kindly request that the authors address them during the revision:

1. Given the existence of several new antibody structure predictors, why was ABodyBuilder2 exclusively selected?

We thank the reviewer for this question, also highlighted by Reviewer 1. We selected ABodyBuilder2 (ABB2) as a recent benchmark (10.1038/s42003-023-04927-7) has shown that it outperforms other structure predictors such as IgFold and EquiFold, especially over the CDRH3 region, which is the most challenging to model. ABB2’s performance is on par with AlphaFold Multimer, however ABB2 takes a fraction of the time to complete each model. This is crucial when generating models of tens of thousands of human antibodies as a reference set and is appropriate for preclinical drug discovery, when many clones remain potential candidates.

We have now clarified our rationale for selecting ABodyBuilder2 in the Results section:

“The original Therapeutic Antibody Profiler used the homology modelling tool ABodyBuilder (10.1080/19420862.2016.12057; ‘ABodyBuilder1’, for clarity) for antibody structural modelling. In 2018, this was the state-of-the-art tool for high-throughput antibody modelling. However, recent advances in deep learning have yielded several pretrained *ab initio* structure prediction architectures that can be applied or adapted to the task of rapid antibody/CDR loop modelling (10.1101/2021.10.04.463034, 10.1038/s41467-023-38063-x, (10.1093/bioinformatics/btac016, 10.1038/s42003-023-04927-7). Their average performance has been shown to be consistently higher than that of homology-based antibody modelling methods. Since better models of antibodies should improve the reliability of our developability guidelines, we explored the case for updating our TAP protocol to use a more recent machine learning-based tool (ABodyBuilder2 (10.1038/s42003-023-04927-7)) for 3D structural modelling. We selected ABodyBuilder2 as it has been shown to outperform other

antibody-specific modelling methods (10.1038/s42003-023-04927-7), while being competitive with AlphaFold Multimer at orders of magnitude faster modelling rates.”

2. What were the reasons behind ABodyBuilder's inability to generate models for Basiliximab, Iscalimab, and Netakimab?

ABodyBuilder1 was unable to model these antibodies due to not having templates of sufficient homology in its databases. We have now clarified this at the appropriate point of the manuscript:

[Table 1 Caption]

“Flagging regions across the five TAP developability metrics calculated over the CST_{all} dataset (See Methods), when therapeutics are modelled by ABodyBuilder1 or by ABodyBuilder2. As ABodyBuilder1 could not find sufficiently homologous templates to produce a model for Basiliximab, Iscalimab, and Netakimab, its statistics are calculated over 661/664 CSTs.”

3. Could the authors provide an in-depth explanation of the distinctions among PPC, PNC, and SFvCSP using both versions of ABodyBuilder? Additionally, please rationalize the superiority of the newer version.

We thank the reviewer for this point, also highlighted by Reviewer 1. Regardless of whether ABodyBuilder1 or ABodyBuilder2 is used for modelling, the PSH, PPC, PNC, and SFvCSP metrics calculated over the same sets of CSTs yield very similar distributions; as we mentioned in the results “The ABodyBuilder2-modeled CST_{all} flagging thresholds show high similarity to those obtained by ABodyBuilder1 (Table 1).” This is not surprising, since we deliberately picked metrics that would not be too sensitive to slight inaccuracies/changes in structure. For example, passing a threshold residue surface exposure is sufficient for a residue to be included in the PSH, PPC, PNC, and SFvCSP calculation. We do not weight based on the degree of exposure or sum over every inter-atomic distance (features that are extremely volatile across runs and particularly prone to systematic differences between tools). Scatter plots for each TAP metric calculated over all CSTs modelled either by ABodyBuilder1 or ABodyBuilder2 are shown in Fig. R1 (the new Fig. S1). They show considerable agreement between the two methods on which CSTs sit at the most extreme property values, with some variation which is to be expected when substantially different 3D structures are predicted. Overall, we note that while the trends in property values are medium to strong positive, they do not align with $x=y$, emphasising the need to be consistent in using the same tool for setting the guidelines and evaluating unseen candidates.

Fig R1. Scatterplots showing the degree of consistency in TAP metric values and thresholds over all CSTs when evaluated on an ABodyBuilder1 (x-axis) or ABodyBuilder2 (y-axis) model. Amber thresholds based on the 5th and/or 95th percentile values for each modelling tool are shown with dashed lines. A least-squares line of best fit for each metric is shown in black, with $x=y$ in grey.

In principle, we should base our TAP guideline thresholds on 3D models generated with state-of-the-art accuracy, a point we highlighted in the Results section: “Since better models of antibodies should improve the reliability of our developability guidelines, we explored the case for updating our TAP protocol...”. We have demonstrated the improved accuracy of ABodyBuilder2 models over ABodyBuilder1 models (Table S1, and reference 10.1038/s42003-023-04927-7). Moreover, as we covered during the section of guideline benchmarking, ABodyBuilder2’s non-deterministic modelling allows us to sample multiple static side chain states, which provides better quantitative agreement with expensive molecular dynamics simulations than single models. In time, as improvements in high-throughput antibody modelling continue, the TAP metrics should be re-evaluated using the new technology. We note that any guidelines should always be derived using sets of models of CSTs generated using the same modelling algorithm, to mitigate against corruption due to systematic biases (10.1073/pnas.1810576116).

>We have added a new Figure S1, scatterplots of PSH, PPC, PNC, and SFvCSP scores for all CSTs evaluated on ABodyBuilder1 (x-axis) or ABodyBuilder2 (y-axis) models.

>This new figure is referenced in “Benchmarking a New TAP Protocol”: “The ABodyBuilder2-modeled CST_{all} flagging thresholds show high similarity to those obtained by ABodyBuilder1 (Table 1, Fig. S1).

>We have added the following sentence to “Benchmarking a New TAP Protocol”: “This emphasises the need to use the same modelling tool for setting the guidelines and evaluating new candidates.”

4. Could the authors offer supporting data to elucidate why downsampling CSTs with longer CDRH3s resulted in a more robust guideline?

We appreciate this question as it reveals some miscommunications on our part. Firstly, we did not deliberately down-sample CSTs with long CDRH3s, this was a by-product of filtering by a threshold model confidence, since longer CDRH3 loops are in general less likely to be confidently modelled. Secondly, we know from the original ABodyBuilder2 paper (10.1038/s42003-023-04927-7; and previous ML antibody modelling approaches such as ABlooper, 10.1093/bioinformatics/btac016) that confidence filtering yields a subset of models with a higher mean accuracy. In this sense, any guidelines established based on a more accurate set of models ought to be closer to the ground truth. However, these guidelines would not be useful for CSTs that cannot be confidently modelled if the physicochemical properties of such antibodies sit in a different distribution solely because of their poorer average modelling accuracy. The fact that the confidence-filtered guidelines align strongly with those set across all CSTs demonstrates that our broader methodology is robust to variable prediction accuracy and is generally compatible with any input candidate of interest.

We have revised the results section of the paper to make our point clearer.

“To investigate the impact of FAPE-based confidence filtering on our guidelines, we first determined an appropriate CDRH3 root-mean squared predicted error threshold that would filter out the least-confidently modeled CDRH3s (1.31 Å, see Methods for the derivation), then calculated our developability guidelines based only on the subset of most confident CST predictions (the CST_{conf} set, see Fig. S1, Table S3). This set of generally higher quality models provides a more accurate reference set of physicochemical distributions, which, if they were to differ substantially from the general set, would imply that ABodyBuilder2's model accuracy has a significant systematic impact on the aggregate guidelines set over all CSTs. Overall, the guidelines derived from only the most confident models aligned closely with those set over all CSTs, suggesting that the new TAP guidelines are robust to the variable prediction accuracy of ABodyBuilder2.”

5. In Fig. 1A and F, the terms "IMGT CDR Length" and "CDR Length" were both employed. Could the authors clarify whether these terms are interchangeable in this context? If so, it would be advisable to use just one term consistently throughout the manuscript.

We thank the reviewer for spotting this. We used the IMGT definition throughout this project; the two terms are interchangeable. To avoid confusion, we now only use the phrase “Total CDR Length” in our figures. The fact that we are using the IMGT definition is captured in our Methods section.

We have amended the x-axis label in Manuscript Figures 1, 3, and 4, and SI Figures S1, S7-S10.

6. While the authors state that CSTs and natural antibodies differ in key physicochemical properties, Fig. 1C - E seems to depict a significant degree of similarity between them. Could the authors provide explanations for this apparent contradiction?

We are grateful for this point. The “key physicochemical properties” we are referring to here

are the CDR length and PSH distributions; their charge properties are similar. We conveyed this in the results section: “CSTs and natural human antibodies adopted similar PPC, PNC, and SFvCSP distributions, but natural antibodies were even more enriched at longer CDRs and higher PSH scores than observed previously”, and in the discussion section: “...the broad trends in property distribution between CSTs and natural antibodies have remained consistent; i.e. CSTs as a whole have significantly shorter CDR loops and smaller patches of surface hydrophobicity, while their charge properties are highly similar.”

We have now revised the phrase “key physicochemical properties” to clarify our meaning.

“In summary, our investigations strengthen the evidence that CSTs and natural antibodies differ in their CDR length and surface hydrophobicity properties.”

--

Reviewer #3 (Remarks to the Author):

Raybould et al. have conducted comprehensive analyses of lambda and kappa properties among clinical-stage therapeutic and natural antibodies, identifying biophysical properties and positions/residues that associate with antibody risk profiles and developability. They build these analyses into their existing tool framework, alongside updates to their machine learning models. Specifically, through this implementation they expand the functionality of their “Therapeutic Antibody Profiler” software, now offering the potential to refine and expand the inclusion of candidate antibodies in initial stages of development – this represents a great resource for the community. The manuscript is thorough and very nicely written. Barring some minor editorial comments, I would be willing to recommend this manuscript for publication in Communications Biology.

We thank the reviewer for their endorsement of our manuscript and for their kind words.

Minor Comments

1) Page 2, paragraph 2: It would probably be good to describe the overall composition of the antibody first (i.e., is composed of two identical heavy and light chains).

We agree and have revised this paragraph accordingly.

“Conventional antibodies are dimeric, comprise two identical heavy and light chains and accomplish precise antigen recognition through two dedicated antigen binding sites, termed the variable regions (Fvs). These Fvs are identical and structurally/chemically intricate, containing six proximal complementarity-determining region (CDR) loops — three on the variable domain of the heavy chain (VH, CDRH1-3) and three on the variable domain of the light chain (VL, CDRL1-3).”

2 & 4) Page 2, paragraph 3: “VH” and “VL” are “encoded” by many genes of different types, which reside in several loci across the genome. If you’re referring to human in this instance, you should articulate the details, perhaps combining this description with that in the following paragraph. Page 2, Paragraph 4: Should specify that you’re discussing human, and a reference or two might be useful.

We thank the reviewer for highlighting the lack of clarity and potential for misunderstanding in these paragraphs. We were referring specifically to human genes in both paragraphs but had not specified this. We have taken the reviewer's advice to integrate the concepts in paragraphs 3 and 4 for a more coherent explanation of antibody genetics, have clarified that we are using human chromosomes as an example, and have added an extra reference.

“A large portion of the VH sequence derives from the recombination of a heavy V, D, and J gene, while most of the VL sequence is analogously the product of recombination of a light V and J gene. These heavy and light chain immunoglobulin ‘germline’ genes are encoded at different loci across the chromosomes. For example, in humans, the heavy chain V, D, and J genes (IGHV, IGHD, IGHJ) lie solely on chromosome 14, while light chain V and J genes exist at two loci; a ‘kappa’ (κ , IGKV and IGKJ) locus on chromosome 2, and a ‘lambda’ (λ , IGLV and IGLJ) locus on chromosome 22 (10.1093/nar/gki010, 10.1002/0471142735.ima01ps40).

Within each locus, different V(D)J genes recombine to create a considerable baseline diversity in both VH and VL sequence (10.1080/19420862.2020.172968). Nucleotide insertion/deletions in the junction region between genes (which falls within the CDR3 loops) further contribute to exceptional VH and VL sequence diversification. Pairing of the recombined heavy and light chains then adds an additional combinatorial diversity; in this manuscript we term antibodies containing a κ light chain as κ -antibodies, and those containing a λ light chain as λ -antibodies. Finally, antibody sequence diversity is magnified through somatic hypermutation during an immune response. This process is often artificially mimicked during therapeutic development through *in vitro* affinity maturation/engineering.”

3) Page 2, paragraph 3: V(D)J recombination is not “largely random”. This is an antiquated description of this process and should be edited.

We thank the reviewer for highlighting this, and fully agree that factors such as chromosome proximity have been shown to bias the likelihood of recombination events. We have removed this phrase from the manuscript.

5) Page 2, Introduction: Use of “VL” vs. “light chain” is inconsistent. (this is also true throughout the manuscript.

*We thank the reviewer for drawing this to our attention. We have carefully revised the manuscript to use “VL” specifically when we are referring to the **variable** domain of the light chain, and “**light chain**” for when we are referring to the entire light chain (**variable and constant** domains). All revised labels are highlighted in the tracked changes.*

6) Page 2, “the same light chain gene origins”: This needs to be clearer (“gene origins” is vague). Do you mean the same germline genes are used consistently?

We appreciate the reviewer pointing this out. We did mean consistent use of the same germline genes and have modified the text:

“Amongst the thousands of anti-coronavirus antibodies independently isolated throughout the pandemic, the same VL germline genes have been frequently observed amongst antibodies with a high confidence of engaging the same epitope.”

7) To improve ease of reading (without needing to review other publications resources), it might be useful to give a little more description of the Nat database used from OAS. Just a very brief description of how that dataset is curated would suffice. This would help more naïve readers understand what is being incorporated into the analysis with CST antibodies.

We are grateful to the reviewer for highlighting this to facilitate the approachability of the paper. We have added some text to the start of the results to help orient the reader:

“We first curated the latest set of non-redundant, post Phase-I clinical stage therapeutics (CSTs) designated for use in humans from Thera-SAbDab (10.1093/nar/gkz827) (25th January, 2023). We obtained 664 CST Fv sequences (the ‘CST_{all}’ dataset), compared to the 242 used in our previous analysis (10.1002/pro.4205). To obtain a reference set of natural human antibodies, we utilised the paired Observed Antibody Space (OAS) database (10.1002/pro.4205), which tracks, cleans, and annotates single-cell antibody V(D)J sequencing datasets in the public domain. We curated all 79,759 non-redundant natively paired human antibody sequences from paired OAS (25th January, 2023), which we term the ‘Nat_{all}’ dataset. This compares to datasets of between 14,000-19,000 artificially-paired human antibody sequences used in our previous analysis (10.1073/pnas.1810576116).”

REVIEWERS' COMMENTS:

Reviewer #1 (Remarks to the Author):

The authors have addressed all of my comments.

Reviewer #2 (Remarks to the Author):

The authors addressed all my comments and concerns.

Reviewer #3 (Remarks to the Author):

Appreciate the authors efforts to revise their manuscript as per the comments of the reviewers.